# WaveFlow: A Compact Flow-based Model for Raw Audio

## Abstract

In this work, we present WaveFlow, a small-footprint generative flow for raw audio, which is trained with maximum likelihood without density distillation and auxiliary losses as used in Parallel WaveNet. It provides a unified view of flow-based models for raw audio, including autoregressive flow (e.g., WaveNet) and bipartite flow (e.g., WaveGlow) as special cases. We systematically study these likelihood-based generative models for raw waveforms in terms of test likelihood and speech fidelity. We demonstrate that WaveFlow can synthesize high-fidelity speech and obtain comparable likelihood as WaveNet, while only requiring a few sequential steps to generate very long waveforms. In particular, our small-footprint WaveFlow has 5.91M parameters and can generate 22.05kHz high-fidelity speech $42.6\times$ faster than real-time on a GPU without engineered inference kernels. [1]

## 1 Introduction

Deep generative models have obtained noticeable successes for modeling raw audio in high-fidelity speech synthesis and music generation (e.g., van den Oord et al., 2016; Dieleman et al., 2018). Autoregressive models are among the best performing generative models for raw audio waveforms, providing the highest likelihood scores and generating high quality samples (e.g., van den Oord et al., 2016; Kalchbrenner et al., 2018). One of the most successful examples is WaveNet (van den Oord et al., 2016), an autoregressive model for waveform synthesis. It operates at the high temporal resolution of raw audio (e.g., 24kHz) and sequentially generates waveform samples at inference. As a result, WaveNet is prohibitively slow for speech synthesis and one has to develop highly engineered kernels for real-time inference (Arık et al., 2017a; Pharris, 2018). [2]

Flow-based models (Dinh et al., 2014; Rezende and Mohamed, 2015) are a family of generative models, in which a simple initial density is transformed into a complex one by applying a series of invertible transformations. One group of models are based on *autoregressive transformation*, including autoregressive flow (AF) and inverse autoregressive flow (IAF) as the "dual" of each other (Kingma et al., 2016; Papamakarios et al., 2017; Huang et al., 2018). AF is analogous to autoregressive models, which performs parallel density evaluation and sequential synthesis. In contrast, IAF performs parallel synthesis but sequential density evaluation, making likelihood-based training very slow. Parallel WaveNet (van den Oord et al., 2018) distills an IAF from a pretrained autoregressive WaveNet, which gets the best of both worlds. However, it requires the density distillation with Monte Carlo approximation and a set of auxiliary losses for good performance, which complicates the training pipeline and increases the cost of development. Instead, ClariNet (Ping et al., 2019) simplifies the density distillation by computing a regularized KL divergence in closed-form.

Another group of flow-based models are based on *bipartite transformation* (Dinh et al., 2017; Kingma and Dhariwal, 2018), which provide parallel density evaluation and parallel synthesis. Most recently, WaveGlow (Prenger et al., 2019) and FloWaveNet (Kim et al., 2019) successfully applies Glow (Kingma and Dhariwal, 2018) and RealNVP (Dinh et al., 2017) for waveform synthesis, respectively. However, the bipartite transformations are less expressive than the autoregressive transformations (see Section 2.3 for detailed discussion). In general, these bipartite flows require

---

[1] Audio samples are located at: `https://waveflow-demo.github.io/`.

[2] Real-time inference is a requirement for most production text-to-speech systems. For example, if the system can synthesize 1 second of speech in 0.5 seconds, it is $2\times$ faster than real-time.

deeper layers, larger hidden size, and huge number of parameters to reach comparable capacities as autoregressive models. For example, WaveGlow and FloWaveNet have 87.88M and 182.64M parameters with 96 layers and 256 residual channels, respectively. In contrast, a 30-layer WaveNet has only 4.57M parameters with 128 residual channels.

In this work, we present WaveFlow, a compact flow-based model for raw audio. Specifically, we make the following contributions:

1. WaveFlow is trained with maximum likelihood without density distillation and auxiliary losses used in Parallel WaveNet (van den Oord et al., 2018) and ClariNet (Ping et al., 2019), which simplifies the training pipeline and reduces the cost of development.

2. WaveFlow squeezes the 1-D raw waveforms into a 2-D matrix and produces the whole audio within a fixed sequential steps. It also provides a unified view of flow-based models for raw audio and allows us to explicitly trade inference efficiency for model capacity. We implement WaveFlow with a dilated 2-D convolutional architecture (Yu and Koltun, 2015), and it includes both Gaussian WaveNet (Ping et al., 2019) and WaveGlow (Prenger et al., 2019) as special cases.

3. We systematically study the likelihood-based generative models for raw audios in terms of test likelihood and speech quality. We demonstrate that WaveFlow can obtain comparable likelihood and synthesize high-fidelity speech as WaveNet (van den Oord et al., 2016), while only requiring a few sequential steps to generate very long waveforms.

4. Our small-footprint WaveFlow has only 5.91M parameters and synthesizes 22.05 kHz high-fidelity speech (MOS: 4.32) more than $40\times$ faster than real-time on a Nvidia V100 GPU. In contrast, WaveGlow (Prenger et al., 2019) requires 87.8M parameters for generating high-fidelity speech. The small memory footprint is preferred in production TTS systems, especially for on-device deployment.

We organize the rest of the paper as follows. Section 2 reviews the flow-based models with autoregressive and bipartite transformations. We present WaveFlow in Section 3 and discuss related work in Section 4. We report experimental results in Section 5 and conclude the paper in Section 6.

## 2 FLOW-BASED GENERATIVE MODELS

Flow-based models (Dinh et al., 2014; 2017; Rezende and Mohamed, 2015) transform a simple density of latent variables $p(\boldsymbol{z})$ (e.g., isotropic Gaussian) into a complex data distribution $p(\boldsymbol{x})$ by applying a bijection $\boldsymbol{x} = f(\boldsymbol{z})$, where $\boldsymbol{x}$ and $\boldsymbol{z}$ are both $n$-dimensional. The probability density of $\boldsymbol{x}$ can be obtained through the change of variables formula:

$$p(\boldsymbol{x}) = p(\boldsymbol{z}) \left| \det \left( \frac{\partial f^{-1}(\boldsymbol{x})}{\partial \boldsymbol{x}} \right) \right|, \tag{1}$$

where $\boldsymbol{z} = f^{-1}(\boldsymbol{x})$ is the inverse transformation, and $\det \left( \frac{\partial f^{-1}(\boldsymbol{x})}{\partial \boldsymbol{x}} \right)$ is the determinant of its Jacobian. In general, it takes $O(n^3)$ to compute the determinant, which is not scalable to high-dimensional data. There are two notable groups of flow-based models with triangular Jacobians and tractable determinants. They are based on autoregressive and bipartite transformations, respectively.

### 2.1 AUTOREGRESSIVE TRANSFORMATION

The autoregressive flow (AF) and inverse autoregressive flow (IAF) (Kingma et al., 2016; Papamakarios et al., 2017) use autoregressive transformations. Specifically, AF defines the inverse transformation $\boldsymbol{z} = f^{-1}(\boldsymbol{x}; \boldsymbol{\vartheta})$ as:

$$z_t = x_t \cdot \sigma_t(x_{<t}; \boldsymbol{\vartheta}) + \mu_t(x_{<t}; \boldsymbol{\vartheta}), \tag{2}$$

where the shifting variables $\mu_t(x_{<t}; \boldsymbol{\vartheta})$ and scaling variables $\sigma_t(x_{<t}; \boldsymbol{\vartheta})$ are modeled by an autoregressive architecture parameterized by $\boldsymbol{\vartheta}$ (e.g., WaveNet). Note that, the $t$-th variable $z_t$ only depends on $x_{\leq t}$, thus the Jacobian is a triangular matrix as illustrated in Figure 1(a) and its determinant is the product of the diagonal entries: $\det \left( \frac{\partial f^{-1}(\boldsymbol{x})}{\partial \boldsymbol{x}} \right) = \prod_t \sigma_t(x_{<t}; \boldsymbol{\vartheta})$. The density $p(\boldsymbol{x})$ can be

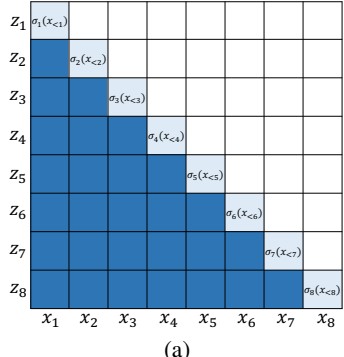 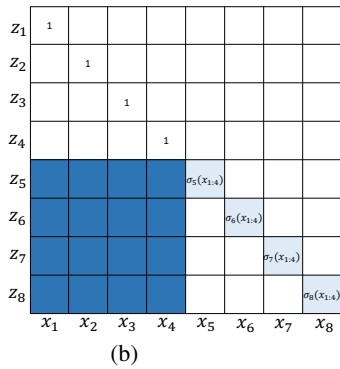

(a)                    (b)

Figure 1: The Jacobian $\frac{\partial f^{-1}(\boldsymbol{x})}{\partial \boldsymbol{x}}$ of (a) an autoregressive transformation, and (b) a bipartite transformation. The blank cells are 0s and represent the independent relations between $z_i$ and $x_j$. The light-blue cells are scaling variables and represent the linear dependencies between $z_i$ and $x_i$. The dark-blue cells represent complex non-linear dependencies defined by neural networks.

easily evaluated by change of variables formula, because $\boldsymbol{z} = f^{-1}(\boldsymbol{x})$ can be computed in parallel from Eq. (2) (i.e., the required $O(n)$ operations can be done in $O(1)$ time on modern GPU hardware). However, AF has to do sequential synthesis, because the forward transformation $\boldsymbol{x} = f(\boldsymbol{z})$ is autoregressive: $x_t = \frac{z_t - \mu_t(x_{<t};\boldsymbol{\vartheta})}{\sigma_t(x_{<t};\boldsymbol{\vartheta})}$. In contrast, IAF uses an autoregressive transformation for $\boldsymbol{z} = f^{-1}(\boldsymbol{x})$:

$$z_t = \frac{x_t - \mu_t(z_{<t};\boldsymbol{\vartheta})}{\sigma_t(z_{<t};\boldsymbol{\vartheta})}, \tag{3}$$

making density evaluation impractically slow for training, but it can do parallel synthesis by $x_t = z_t \cdot \sigma_t(z_{<t};\boldsymbol{\vartheta}) + \mu_t(z_{<t};\boldsymbol{\vartheta})$. Parallel WaveNet (van den Oord et al., 2018) and ClariNet (Ping et al., 2019) are based on IAF, which lacks efficient density evaluation and relies on distillation from a pretrained autoregressive WaveNet.

## 2.2 BIPARTITE TRANSFORMATION

RealNVP (Dinh et al., 2017) and Glow (Kingma and Dhariwal, 2018) use bipartite transformation by partitioning the data $\boldsymbol{x}$ into two groups $\boldsymbol{x}_a$ and $\boldsymbol{x}_b$, where the indices sets $a \cup b = \{1, \cdots, n\}$ and $a \cap b = \phi$. Then, the inverse transformation $\boldsymbol{z} = f^{-1}(\boldsymbol{x}, \boldsymbol{\theta})$ is defined as:

$$z_a = x_a, \quad z_b = x_b \cdot \sigma_b(x_a;\boldsymbol{\theta}) + \mu_b(x_a;\boldsymbol{\theta}). \tag{4}$$

where the shifting variables $\mu_b(x_a;\boldsymbol{\theta})$ and scaling variables $\sigma_b(x_a;\boldsymbol{\theta})$ are modeled by a feed-forward neural network. The Jacobian $\frac{\partial f^{-1}(\boldsymbol{x})}{\partial \boldsymbol{x}}$ is a special triangular matrix as illustrated in Figure 1 (b). By definition, the forward transformation $\boldsymbol{x} = f(\boldsymbol{z}, \boldsymbol{\theta})$ is,

$$x_a = z_a, \quad x_b = \frac{z_b - \mu_b(x_a;\boldsymbol{\theta})}{\sigma_b(x_a;\boldsymbol{\theta})}, \tag{5}$$

and can also be done in parallel. As a result, the bipartite transformation provides both parallel density evaluation and parallel synthesis. In previous work, WaveGlow (Prenger et al., 2019) and FloWaveNet (Kim et al., 2019) both squeeze the adjacent audio samples on the channel dimension, and apply the bipartite transformation on the partitioned channel dimension.

## 2.3 CONNECTIONS

It is worthwhile to mention that the autoregressive transformation is more expressive than bipartite transformation in general. As illustrated in Figure 1(a) and (b), the autoregressive transformation introduces $\frac{n \times (n-1)}{2}$ complex non-linear dependencies (dark-blue cells) and $n$ linear dependencies between data $\boldsymbol{x}$ and latents $\boldsymbol{z}$. In contrast, bipartite transformation introduces only $\frac{n^2}{4}$ non-linear

dependencies and $\frac{n}{2}$ linear dependencies. Indeed, one can reduce an autoregressive transformation $\boldsymbol{z} = f^{-1}(\boldsymbol{x}; \boldsymbol{\vartheta})$ to a bipartite transformation $\boldsymbol{z} = f^{-1}(\boldsymbol{x}; \boldsymbol{\theta})$ by: (i) picking an autoregressive order $\boldsymbol{o}$ such that all of the indices in set $a$ rank early than the indices in $b$, and (ii) setting the shifting and scaling variables as,

$$\mu_t(x_{<t}; \boldsymbol{\vartheta}) = \begin{cases} 0 & \text{for } t \in a \\ \mu_t(x_a; \boldsymbol{\theta}) & \text{for } t \in b \end{cases}, \quad \sigma_t(x_{<t}; \boldsymbol{\vartheta}) = \begin{cases} 1 & \text{for } t \in a \\ \sigma_t(x_a; \boldsymbol{\theta}) & \text{for } t \in b \end{cases}.$$

Given the less expressive building block, the bipartite transformation-based flows generally require many more layers and larger hidden size to match the capacity of a compact autoregressive models (e.g., as measured by test likelihood) (Kingma and Dhariwal, 2018; Prenger et al., 2019).

## 3 WaveFlow

In this section, we present WaveFlow and its implementation with dilated 2-D convolutions.

### 3.1 Definition

We denote the high dimensional 1-D waveform as $\boldsymbol{x} = \{x_1, \cdots, x_n\}$. We first *squeeze* $\boldsymbol{x}$ into a $h$-row 2-D matrix $X \in \mathbb{R}^{h \times w}$ by column-major order, where $w = \frac{n}{h}$ and adjacent samples are in the same column. We assume $Z \in \mathbb{R}^{h \times w}$ are sampled from an isotropic Gaussian, and define the inverse transformation $Z = f^{-1}(X; \Theta)$ as,

$$Z_{i,j} = \sigma_{i,j}(X_{<i,\bullet}; \Theta) \cdot X_{i,j} + \mu_{i,j}(X_{<i,\bullet}; \Theta), \tag{6}$$

where $X_{<i,\bullet}$ represents all elements above $i$-th row (see Figure 2 for an illustration). Note that, i) the receptive fields over the squeezed inputs $X$ for computing $Z_{i,j}$ in WaveFlow is strictly larger than that of WaveGlow when $h > 2$. ii) WaveNet is equivalent to an autoregressive flow with column-major order on the squeezed inputs $X$. iii) Both WaveFlow and WaveGlow look at future waveform samples in original $\boldsymbol{x}$ for computing $Z_{i,j}$, whereas WaveNet can not. iv) The autoregressive flow with row-major order has larger receptive fields than WaveFlow and WaveGlow.

The shifting variables $\mu_{i,j}(X_{<i,\bullet}; \Theta)$ and scaling variables $\sigma_{i,j}(X_{<i,\bullet}; \Theta)$ in Eq. (6) are modeled by a 2-D convolutional neural network detailed in Section 3.2. By definition, the variable $Z_{i,j}$ only depends on the current $X_{i,j}$ and previous $X_{<i,\bullet}$ in raw-major order, thus the Jacobian is a triangular matrix and its determinant is:

$$\det\left(\frac{\partial f^{-1}(X)}{\partial X}\right) = \prod_{i=1}^{h} \prod_{j=1}^{w} \sigma_{i,j}(X_{<i,\bullet}; \Theta). \tag{7}$$

As a result, the log-likelihood can be calculated in parallel by change of variable formula in Eq. (1),

$$\log p(X) = -\sum_{i=1}^{h} \sum_{j=1}^{w} \left(Z_{i,j}^2 + \frac{1}{2}\log(2\pi)\right) + \sum_{i=1}^{h} \sum_{j=1}^{w} \log \sigma_{i,j}(X_{<i,\bullet}; \Theta), \tag{8}$$

and one can do maximum likelihood training efficiently. At synthesis, one may first sample $Z$ from the isotropic Gaussian and apply the forward transformation $X = f(Z; \Theta)$:

$$X_{i,j} = \frac{Z_{i,j} - \mu_{i,j}(X_{<i,\bullet}; \Theta)}{\sigma_{i,j}(X_{<i,\bullet}; \Theta)}, \tag{9}$$

which is only autoregressive on height dimension. Thus, it requires $h$ sequential steps to generate the whole waveform $X$. In practice, a small $h$ (e.g., 8 or 16) works well, thus we can generate very long waveforms within a few sequential steps.

### 3.2 Implementation with dilated 2-D convolutions

In this work, we implement WaveFlow with a dilated 2-D convolutional architecture. Specifically, we use a stack of 2-D convolution layers (e.g., 8 layers in all experiments) to model the shifting variables $\mu_{i,j}(X_{<i,\bullet}; \Theta)$ and scaling variables $\sigma_{i,j}(X_{<i,\bullet}; \Theta)$ in Eq. (6). We use the similar architecture as

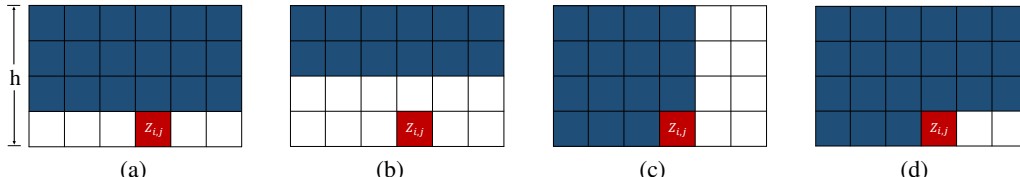

Figure 2: The receptive fields over the squeezed inputs $X$ for computing $Z_{i,j}$ in (a) WaveFlow, (b) WaveGlow, (c) autoregressive flow with column-major order (e.g., WaveNet), and (d) autoregressive flow with row-major order.

Table 1: The test log-likelihoods (LLs) of WaveFlow with different dilation cycles on the height dimension. Both models are stacked with 8 flows and each flow has 8 convolutional layers.

| Model | res. channels | dilations $d$ | receptive field $r$ | test LLs |
|---|---|---|---|---|
| WaveFlow ($h = 32$) | 128 | $1, 1, 1, 1, 1, 1, 1, 1$ | 17 | 4.960 |
| WaveFlow ($h = 32$) | 128 | $1, 2, 4, 1, 2, 4, 1, 2$ | 35 | 5.055 |

Table 2: The heights and corresponding dilations used in our experiments. Note that, the receptive fields are only slightly larger than height $h$.

| Height $h$ | filter size $k$ | dilations $d$ | receptive field $r$ |
|---|---|---|---|
| 8 | 3 | $1, 1, 1, 1, 1, 1, 1, 1$ | 17 |
| 16 | 3 | $1, 1, 1, 1, 1, 1, 1, 1$ | 17 |
| 32 | 3 | $1, 2, 4, 1, 2, 4, 1, 2$ | 35 |
| 64 | 3 | $1, 2, 4, 8, 16, 1, 2, 4$ | 77 |

WaveNet (van den Oord et al., 2016) by replacing the dilated 1-D convolution to 2-D convolution (Yu and Koltun, 2015), while still keeping the gated-tanh nonlinearities, residual connections and skip connections.

We set the filter sizes as 3 for both height and width dimensions. We use non-causal convolutions on width dimension and set the dilation cycle as $[1, 2, 4, \cdots, 2^7]$. The convolutions on height dimension are causal with an autoregressive constraint, and their dilation cycle needs to be designed carefully. In practice, we find the following rules of thumb are important to obtain good results:

- As motivated by the dilation cycle of WaveNet (van den Oord et al., 2016), the dilations of 8 layers should be set as $d = [1, 2, \cdots, 2^s, 1, 2, \cdots, 2^s, \cdots]$, where $s \leq 7$. [3]
- The receptive field $r$ over the height dimension should be larger than the squeezed height $h$. Otherwise, it explicitly introduces unnecessary conditional independence and leads to lower likelihood (see Table 1 for an example). Note that, the receptive field of a stack of dilated convolutional layers is: $r = (k-1) \times \sum_i d_i + 1$, where $k$ is the filter size and $d_i$ is the dilation at $i$-th layer. Thus, the sum of dilations should satisfy: $\sum_i d_i \geq \frac{h-1}{k-1}$. However, when $h$ is larger than or equal to $2^8 = 512$, we simply set the dilation cycle as $[1, 2, 4, \cdots, 2^7]$.
- When the receptive field $r$ has already been larger than $h$, we find that convolutions with smaller dilation and fewer holes provide larger likelihood.

We summarize the heights and preferred dilations in our experiments in Table 2. Note that, WaveFlow becomes fully autoregressive when we squeeze $x$ by its length (i.e. $h = n$) and set its filter size as 1 over the width dimension, which is equivalent to a Gaussian WaveNet learned by MLE (Ping et al., 2019). If we squeeze $x$ by $h = 2$ and set the filter size as 1 on the height dimension, WaveFlow becomes a bipartite flow and is equivalent to WaveGlow with squeezed channels 2.

## 3.3 CONDITIONAL GENERATION

In neural speech synthesis, a neural vocoder (e.g., WaveNet) synthesizes the time-domain waveforms. It can be conditioned on linguistic features (van den Oord et al., 2016; Arık et al., 2017a), the mel-spectrograms from a text-to-spectrogram model (Ping et al., 2018; Shen et al., 2018), or the

---

[3]We did try different setups, but they all lead to worse likelihood scores.

Table 3: The test LLs of WaveFlow with different permutation strategies. All models consist of 8 flows and each flow has 8 convolutional layers with filter sizes 3.

| Model | res. channels | permutation strategy | test LLs |
|---|---|---|---|
| WaveFlow ($h = 16$) | 64 | none | 4.551 |
| WaveFlow ($h = 16$) | 64 | (i) 8 reverse | 4.954 |
| WaveFlow ($h = 16$) | 64 | (ii) 4 reverse + 4 split & reverse | 4.971 |

learned hidden representation within a text-to-wave architecture (Ping et al., 2019). In this work, we test WaveFlow by conditioning it on ground truth mel-spectrograms as in previous work (Prenger et al., 2019; Kim et al., 2019). The mel-spectrogram is upsampled to the same resolution as waveform samples by transposed 2-D convolutions (Ping et al., 2019). To aligned with the squeezed waveform, they are squeezed to the shape $c \times h \times w$, where $c$ is the feature dimension (e.g, bands of the spectrogram). After a $1 \times 1$ convolution mapping the features to residual channels, they are added as the bias term at each layer (van den Oord et al., 2016).

### 3.4    STACKING MULTIPLE FLOWS WITH PERMUTATIONS OVER HEIGHT DIMENSION

Flow-based models require a series of transformations until the distribution $p(X)$ reaches a desired level of complexity (e.g., Rezende and Mohamed, 2015). We let $X = Z^{(n)}$ and repeatedly apply the transformation $Z^{(i-1)} = f^{-1}(Z^{(i)}; \Theta^{(i)})$ defined in Eq. (6) from $Z^{(n)} \to \dots Z^{(i)} \to \dots Z^{(0)}$. We assume $Z^{(0)}$ is from the isotropic Gaussian distribution. The likelihood $p(X)$ can be evaluated by iteratively applying the chain rule:

$$p(X) = p(Z^{(0)}) \prod_{i=1}^{n} \left| \det \left( \frac{\partial f^{-1}(Z^{(i)}; \Theta^{(i)})}{\partial Z^{(i)}} \right) \right|.$$

We find that permuting each $Z^{(i)}$ over the height dimension after each transformation can significantly improve the likelihood scores. In particular, we test two permutation strategies for WaveFlow models stacked with 8 flows (i.e., $X = Z^{(8)}$) in Table 3: (i) we reverse each $Z^{(i)}$ over the height dimension after each transformation, and (ii) we reverse $Z^{(7)}$, $Z^{(6)}$, $Z^{(5)}$, $Z^{(4)}$ over the height dimension as before, but split $Z^{(3)}$, $Z^{(2)}$, $Z^{(1)}$, $Z^{(0)}$ in the middle of the height dimension then reverse each part respectively. [4] Note that, one also needs to permute the conditioner on the height dimension accordingly, which is aligned with $Z^{(i)}$. From Table 3, both (i) and (ii) significantly outperform the model without permutations mainly because of bidirectional modeling. Strategy (ii) outperforms (i) because of its diverse autoregressive orders.

## 4    RELATED WORK

Deep neural networks for speech synthesis (a.k.a. text-to-speech) have received a lot of attention. Over the past few years, several neural text-to-speech (TTS) systems have been introduced, including WaveNet (van den Oord et al., 2016), Deep Voice (Arık et al., 2017a), Deep Voice 2 (Arık et al., 2017b), Deep Voice 3 (Ping et al., 2018), Tacotron (Wang et al., 2017), Tacotron 2 (Shen et al., 2018), Char2Wav (Sotelo et al., 2017), VoiceLoop (Taigman et al., 2018), WaveRNN (Kalchbrenner et al., 2018), ClariNet (Ping et al., 2019), Transformer TTS (Li et al., 2019), ParaNet (Peng et al., 2019) and FastSpeech (Ren et al., 2019).

Neural vocoders, such as WaveNet, play the most important role in recent advances of speech synthesis. In previous work, the state-of-the-art neural vocoders are autoregressive models (van den Oord et al., 2016; Mehri et al., 2017; Kalchbrenner et al., 2018). Several engineering endeavors have been advocated for speeding up their sequential generation process (Arık et al., 2017a; Kalchbrenner et al., 2018). In particular, Subscale WaveRNN (Kalchbrenner et al., 2018) folds a long waveform sequence $x_{1:n}$ into a batch of shorter sequences and can produces up to 16 samples per step, thus it requires at least $\frac{n}{16}$ steps to generate the whole audio. Note that, this is different from the proposed WaveFlow, which can generate $x_{1:n}$ within a fixed number of steps (e.g., 16). Most recently, flow-based models have been successfully applied for parallel waveform synthesis with comparable fidelity

---

[4]After split & reverse operations, the height dimension $[0, \cdots, \frac{h}{2} - 1, \frac{h}{2}, \cdots, h - 1]$ becomes $[\frac{h}{2} - 1, \cdots, 0, h - 1, \cdots, \frac{h}{2}]$.

as autoregressive models (van den Oord et al., 2018; Ping et al., 2019; Prenger et al., 2019; Kim et al., 2019; Yamamoto et al., 2019; Serrà et al., 2019). Among these models, WaveGlow (Prenger et al., 2019) and FloWaveNet (Kim et al., 2019) have a simple training pipeline as they solely use the maximum likelihood objective. However, both of them are less expressive than autoregressive models as indicated by their lower likelihood scores.

Flow-based models can either represent the approximate posteriors for variational inference (Rezende and Mohamed, 2015; Kingma et al., 2016; Berg et al., 2018), or can be trained directly on data using the change of variables formula (Dinh et al., 2014; 2017; Kingma and Dhariwal, 2018; Grathwohl et al., 2018). In previous work, Glow (Kingma and Dhariwal, 2018) extends RealNVP (Dinh et al., 2017) with invertible $1 \times 1$ convolution, and can generate high quality images. Later on, Hoogeboom et al. (2019) generalizes the $1 \times 1$ convolution to invertible $d \times d$ convolutions which operate both channel and spatial axes.

## 5 EXPERIMENT

In this section, we compare likelihood-based generative models for raw audio in term of test likelihood, speech quality and synthesis speed.

**Data:** We use the LJ speech dataset (Ito, 2017) containing about 24 hours of audio with a sampling rate of 22.05kHz recorded on a MacBook Pro in a home enviroment. It consists of $13,100$ audio clips of a single female speaker reading passages from 7 non-fiction books.

**Models:** We evaluate several likelihood-based generative models, including Gaussian WaveNet, WaveGlow, WaveFlow and autoregressive flow (AF). As in Section 3.2, we implement autoregressive flow from WaveFlow by squeezing the waveforms by its length and setting the filter size as 1 for width dimension. Both WaveNet and AF have 30 layers with dilation cycle $[1, 2, \cdots, 512]$ and filter size 3. For WaveGlow and WaveFlow, we investigate different setups, including the number of flows, size of residual channels, and squeezed height $h$.

**Conditioner:** We use the 80-band mel-spectrogram of the original audio as the conditioner for WaveNet, WaveGlow, and WaveFlow. We use FFT size 1024, hop size 256, and window size 1024. For WaveNet and WaveFlow, we upsample the mel conditioner 256 times by applying two layers of transposed 2-D convolution (in time and frequency) interleaved with leaky ReLU ($\alpha = 0.4$). The upsampling strides in time are 16 and the 2-D convolution filter sizes are $[32, 3]$ for both layers. For WaveGlow, we directly use the open source implementation. [5]

**Training:** We train all models on 8 Nvidia 1080Ti GPUs using randomly chosen short clips of $16,000$ samples from each utterance. For WaveFlow and WaveNet, we use the Adam optimizer (Kingma and Ba, 2015) with a batch size of 8 and a constant learning rate of $2 \times 10^{-4}$. For WaveGlow, we use the Adam optimizer with a batch size of 16 and a learning rate of $1 \times 10^{-4}$. We applied weight normalization (Salimans and Kingma, 2016) whenever possible.

### 5.1 LIKELIHOOD

The test log-likelihoods (LLs) of all models are evaluate at 1M training steps. Note that, i) all of the LLs decrease slowly after 1M steps and ii) it took one month to train the largest WaveGlow (residual channels = 512) for 1M steps. Thus, we chose 1M as the cut-off to compare these models. We summarize the results in Table 4 with models from row (a) to (t). We draw the following observations:

- Stacking a large number of flows improves LLs for WaveFlow, autoregressive flow, and WaveGlow. For example, (m) WaveFlow with 8 flows provide larger LL than (l) WaveFlow with 6 flows. The (*b*) autoregressive flow obtains the highest likelihood and even outperforms (*a*) WaveNet with the same amount of parameters. Indeed, AF provides bidirectional modeling by stacking 3 flows interleaved with reverse operations.

- WaveFlow has much larger likelihood than WaveGlow with comparable number of parameters. In particular, a small-footprint (*k*) WaveFlow has only 5.91M parameters but can provide comparable likelihood (5.023 vs. 5.026) as the largest (*g*) WaveGlow with 268.29M parameters.

---

[5] https://github.com/NVIDIA/waveglow

Table 4: The test log-likelihoods (LLs) of all models conditioned on mel-spectrograms. For $a \times b = c$ in the **flows**×**layers** column, $a$ is number of flows, $b$ is number of layers in each flow, and $c$ is the total number of layers. In WaveFlow, $h$ is the squeezed height. Models with bolded test LLs are mentioned in the text.

| | Model | flows×layers | res. channels | # param | test LLs |
|---|---|---|---|---|---|
| (a) | WaveNet | $1 \times 30 = 30$ | 128 | 4.57 M | **5.059** |
| (b) | Autoregressive flow | $3 \times 10 = 30$ | 128 | 4.54 M | **5.161** |
| (c) | WaveGlow | $12 \times 8 = 96$ | 64 | 17.59 M | 4.804 |
| (d) | WaveGlow | $12 \times 8 = 96$ | 128 | 34.83 M | 4.927 |
| (e) | WaveGlow | $6 \times 8 = 48$ | 256 | 47.22 M | 4.922 |
| (f) | WaveGlow | $12 \times 8 = 96$ | 256 | 87.88 M | 5.018 |
| (g) | WaveGlow | $12 \times 8 = 96$ | 512 | 268.29 M | **5.026** |
| (h) | WaveFlow ($h = 8$) | $8 \times 8 = 64$ | 64 | 5.91 M | 4.935 |
| (i) | WaveFlow ($h = 16$) | $8 \times 8 = 64$ | 64 | 5.91 M | 4.954 |
| (j) | WaveFlow ($h = 32$) | $8 \times 8 = 64$ | 64 | 5.91 M | 5.002 |
| (k) | WaveFlow ($h = 64$) | $8 \times 8 = 64$ | 64 | 5.91 M | **5.023** |
| (l) | WaveFlow ($h = 8$) | $6 \times 8 = 48$ | 96 | 9.58 M | 4.946 |
| (m) | WaveFlow ($h = 8$) | $8 \times 8 = 64$ | 96 | 12.78 M | 4.977 |
| (n) | WaveFlow ($h = 16$) | $8 \times 8 = 64$ | 96 | 12.78 M | 5.007 |
| (o) | WaveFlow ($h = 16$) | $6 \times 8 = 48$ | 128 | 16.69 M | 4.990 |
| (p) | WaveFlow ($h = 8$) | $8 \times 8 = 64$ | 128 | 22.25 M | 5.009 |
| (q) | WaveFlow ($h = 16$) | $8 \times 8 = 64$ | 128 | 22.25 M | 5.028 |
| (r) | WaveFlow ($h = 32$) | $8 \times 8 = 64$ | 128 | 22.25 M | **5.055** |
| (s) | WaveFlow ($h = 16$) | $6 \times 8 = 48$ | 256 | 64.64 M | 5.064 |
| (t) | WaveFlow ($h = 16$) | $8 \times 8 = 64$ | 256 | 86.18 M | **5.101** |

- As we increase $h$, the likelihood of WaveFlow steadily increases (can be seen from (h)-(k)), and its inference is getting slower with more sequential steps. In the limit, it is equivalent to an autoregressive flow. It illustrates the trade-off between model capacity and inference efficiency.

- (r) WaveFlow with 128 residual channels can obtain comparable likelihood (5.055 vs 5.059) as (a) WaveNet with 128 residual channels. A larger (t) WaveFlow with 256 residual channels can obtain even larger likelihood than WaveNet (5.101 vs 5.059).

## 5.2 SPEECH FIDELITY AND SYNTHESIS SPEED

We train WaveNet for 1M steps. We train WaveGlow and WaveFlow for 2M steps with small residual channels (64, 96 and 128). We train larger models (res. channels 256 and 512) for 1M steps due to the practical time constraint. At synthesis, we sampled $Z$ from an isotropic Gaussian with standard deviation 1.0 and 0.6 (default) for WaveFlow and WaveGlow, respectively. For WaveFlow and WaveGlow, we run synthesis under NVIDIA Apex with 16-bit floating point (FP16) arithmetic, which does not introduce any degradation of audio fidelity and brings about $2\times$ speedup. We use the crowdMOS tookit (Ribeiro et al., 2011) for naturalness evaluation, where test utterances from these models were presented to workers on Mechanical Turk. We also test the synthesis speed on a Nvidia V100 GPU without using any customized inference kernels. We only implement convolution queues (Paine et al., 2016) in Python to cache the intermediate hidden states within WaveFlow for autoregressive inference over the height dimension, which brings about $4\times$ speedup. We use the permutation strategy (ii) described in Section 3.4 for WaveFlow.

We report the 5-scale Mean Opinion Score (MOS), synthesis speed and model footprint in Table 5. We draw the following observations:

- The small WaveFlow (res. channels 64) has 5.91M parameters and can synthesize 22.05 kHz high-fidelity speech (MOS: 4.32) $42.60\times$ faster than real-time. In contrast, the speech quality of small WaveGlow (res. channels 64) is significantly worse (MOS: 2.17). Indeed, WaveGlow (res. channels 256) requires 87.88M parameters for generating high-fidelity speech.

- The large WaveFlow (res. channels 256) outperforms the same size WaveGlow in terms of speech fidelity (MOS: 4.43 vs. 4.34). It also matches the state-of-the-art WaveNet while gener-

Table 5: The synthesis speed over real-time and the 5-scale Mean Opinion Score (MOS) ratings with 95% confidence intervals. We use 30-layer WaveNet, 96-layer WaveGlow, and 64-layer WaveFlow. Models with bolded numbers are mentioned in the text.

| Model | flows×layers | res. channels | # param | syn. speed | MOS |
|---|---|---|---|---|---|
| WaveNet | $1 \times 30 = 30$ | 128 | 4.57 M | 0.002× | $4.43 \pm 0.14$ |
| WaveGlow | $12 \times 8 = 96$ | 64 | 17.59 M | 93.53× | $2.17 \pm 0.13$ |
| WaveGlow | $12 \times 8 = 96$ | 128 | 34.83 M | 69.88× | $2.97 \pm 0.15$ |
| WaveGlow | $12 \times 8 = 96$ | 256 | 87.88 M | 34.69× | $4.34 \pm 0.11$ |
| WaveGlow | $12 \times 8 = 96$ | 512 | 268.29 M | 8.08× | $4.32 \pm 0.12$ |
| WaveFlow ($h = 8$) | $8 \times 8 = 64$ | 64 | 5.91 M | 47.61× | $4.26 \pm 0.12$ |
| WaveFlow ($h = 16$) | $8 \times 8 = 64$ | 64 | **5.91 M** | **42.60×** | $\mathbf{4.32 \pm 0.08}$ |
| WaveFlow ($h = 16$) | $8 \times 8 = 64$ | 96 | 12.78 M | 26.23× | $4.34 \pm 0.13$ |
| WaveFlow ($h = 16$) | $8 \times 8 = 64$ | 128 | 22.25 M | 21.32× | $4.38 \pm 0.09$ |
| WaveFlow ($h = 16$) | $8 \times 8 = 64$ | 256 | 86.18 M | 8.42× | $\mathbf{4.43 \pm 0.10}$ |
| Ground-truth | — | — | — | — | $4.56 \pm 0.09$ |

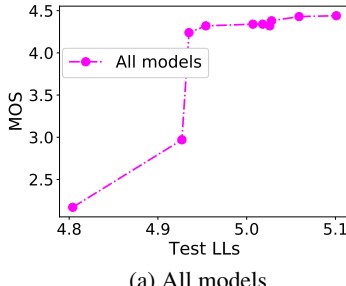

(a) All models

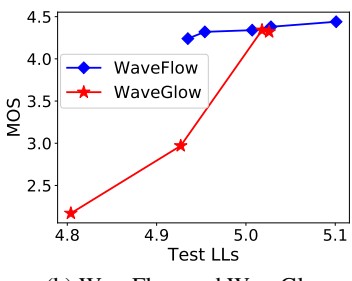

(b) WaveFlow and WaveGlow

Figure 3: The test log-likelihoods (LLs) vs. MOS scores for all likelihood-based models in Table 5.

  ating speech $8.42 \times$ faster than real-time, because it only requires 128 sequential steps (number of flows × height $h$) to synthesize very long waveforms.

- We find a positive correlation between the test likelihoods and MOS scores for these likelihood-based generative models (see Figure 3 for an illustration). One can see that larger LLs correspond to higher MOS scores even when we compare all models in Figure 3 (a). The correlations become more evident, as we consider WaveFlow and WaveGlow in Figure 3 (b).

## 6 CONCLUSION

We propose WaveFlow, a compact flow-based model for raw audio, which can be directly trained with maximum likelihood estimation. It provides a unified view of flow-based models for time-domain waveforms, and includes WaveNet and WaveGlow as special cases. WaveFlow requires a small number of sequential steps to generate high-fidelity speech and obtains likelihood comparable to WaveNet. In the end, our small-footprint WaveFlow can generate 22.05kHz high-fidelity speech more than $40 \times$ faster than real-time on a GPU without engineered inference kernels.

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
