# OpenReview forum: "WaveFlow: A Compact Flow-based Model for Raw Audio"
_ICLR.cc/2020/Conference — Reject_

### Official Review · AnonReviewer3 · 2019-10-23
**Official Blind Review #3**

**Rating:** 8

**Review:**


## Updated review

I have read the rebuttal. The new version of the paper is definitely clearer, especially the contribution section and the experimental results. The new version addresses all my concerns, hence I am upgrading my rating to Accept.

## Original review

This paper presents the WaveGlow model, a generative model for raw audio. The model is based on a 2D-matrix approach, which allows to generate the audio with a fixed amount of step. The model is shown to be a generalization of the two main approaches for raw audio generation, autoregressive flow and bipartite flow. The model is evaluated and compared with related work on an objective evaluation (Log-likelihood) and a subjective evaluation (MOS), and is shown to be a trade-off between memory footprint, generation speed and quality.

I think this paper should be accepted, for the following reasons:
- The theoretical framework presented is novel and significant, as it provides a unified view of the two main approaches for neural waveform generation.
- The experiments are reasonably convincing, although they could be improved.

Detailed comments:
- In the subjective evaluation section (5.2), Table 5 is hard to decipher, especially given that there are three measurements to take into account, so it's not easy to see the benefit of the approach. Maybe the results should be organised differently, for instance grouping them according to one measurement could help, typically showing what speed and MOS each of the three models can achieve for a given model size. Maybe plotting speed vs MOS for the same model size could also be interesting.
- In the same section, is the WaveNet model the original one, or the Parallel WaveNet ? if it's the original, why not include Parallel WaveNet in the table ?
- Typo at the end of Section 1: "We orgnize" -> "organize"

**Experience Assessment:**

I have read many papers in this area.

**Review Assessment: Checking Correctness Of Derivations And Theory:**

I did not assess the derivations or theory.

**Review Assessment: Checking Correctness Of Experiments:**

I carefully checked the experiments.

**Review Assessment: Thoroughness In Paper Reading:**

I read the paper at least twice and used my best judgement in assessing the paper.

---

> ### Author Response · Authors · 2019-11-14
> **Response to Official Blind Review #3**
>
> Many thanks for your detailed review; they are really helpful to improve the quality of our paper.
>
> ** After the submission, we have made two improvements: 1) We find that the split & reverse operations for stacking multiple flows are more effective than reverse-only operations (see details in Section 3.4 in our revision). Our small-footprint WaveFlow now obtains larger test likelihood (see Table 3 in the revision) and improved speech fidelity (see Table 5 in the revision).   2) We implement convolution queue (Paine et al., 2016), which brings additional 3x to 5x speedup for WaveFlow models.  As a result, our small-footprint WaveFlow (5.91M parameters) can now generate 22.05kHz high-fidelity speech (MOS: 4.32) more than 40x faster than real-time (faster than WaveGlow), which provides a very promising neural vocoder. We also recommend setting h = 16 for neural vocoding task, because it provides better speech fidelity and its synthesis speed is only marginally slower than h = 8 with the help of convolution queue. **
>
> We will address your detailed comments in the following.
>
> - “In the subjective evaluation section (5.2), Table 5 is hard to decipher, especially given that there are three measurements to take into account, so it's not easy to see the benefit of the approach.”
> * This is a good point.  In comparison with WaveGlow, WaveFlow requires much fewer parameters (5.91M vs. 87.88M) to generate comparable fidelity speech (MOS: 4.32 vs. 4.34). Its synthesis speed is also slightly faster (42.60x vs. 34.69x).  In comparison with WaveNet, WaveFlow models synthesize speech significantly faster (e.g., 42.60x vs. 0.002x). We will emphasize the benefit of our approach in the final draft. We will also try to organize the results in a clearer way. Many thanks for your nice suggestions.
>
> - “In the same section, is the WaveNet model the original one, or the Parallel WaveNet ? if it's the original, why not include Parallel WaveNet in the table?”
> * It is the autoregressive WaveNet. Note that, reproducing Parallel WaveNet, which produces high fidelity speech on public dataset, is so far beyond the capability of open source community. For example, here is some related discussion ( https://github.com/r9y9/wavenet_vocoder/issues/7 ).
>
> Also, many thanks for pointing out the typo. We have fixed it.

---

### Official Review · AnonReviewer1 · 2019-10-23
**Official Blind Review #1**

**Rating:** 6

**Review:**

This submission belongs to the field of text-to-speech synthesis. In particular it looks at a novel way of formulating a normalising flow using 2D rather than conventional 1D representation. Such reformulation enables to provide interpretations to several existing approaches as well as formulate a new one with quite interesting properties. This submission would benefit from a discussion of limitations of your approach.

I believe there is a great deal of interest in the use of normalising flows in the text-to-speech area. I believe this submission could be a good contribution to the area. The test log-likelihoods look comparable to existing approaches with significantly worse inference times. The mean opinion scores (MOS) seem to approach one of the standard baselines with significantly worse inference times though at the expense of increasing the number of model parameters from 6M to 86M parameters whilst gaining only 0.2 in MOS. The submission would have benefited from discussion about model complexity/expressivity and it's impact on MOS for WaveFlow, WaveNet and other approaches.

The largest issues with this submission are:

1) lack of proper technical description of your model in sections 1 and 2 making reading sections 1,2,3,etc in order awkward. It seems the order should be 3,4,(5),1,2,(5).
2) complete omission of conditioning on text to be synthesised; anyone not familiar deeply with speech synthesis will wonder where does the text come in
3) explicit statement of complexity for the operations involved using proper big-O notation; helps to avoid confusion about what do you mean by "parallel" (autoregressive WaveNet followed by parallel computation != parallel computation)


**Experience Assessment:**

I have published in this field for several years.

**Review Assessment: Checking Correctness Of Derivations And Theory:**

I carefully checked the derivations and theory.

**Review Assessment: Checking Correctness Of Experiments:**

I carefully checked the experiments.

**Review Assessment: Thoroughness In Paper Reading:**

I read the paper thoroughly.

---

> ### Author Response · Authors · 2019-11-14
> **Response to Official Blind Review #1**
>
> Thank you so much for the detailed comments and suggestions; they are really helpful to improve the quality of our paper.
>
> ** After the submission, we have made two improvements: 1) We find that the split & reverse operations for stacking multiple flows are more effective than reverse-only operations (see details in Section 3.4 in the revision). Our small-footprint WaveFlow now obtains larger test likelihood (see Table 3 in our revision) and improved speech fidelity (see Table 5 in the revision).   2) We implement convolution queue (Paine et al., 2016), which brings additional 3x to 5x speedup for WaveFlow models.  As a result, our small-footprint WaveFlow (5.91M parameters) can generate 22.05kHz high-fidelity speech (MOS: 4.32) more than 40x faster than real-time (faster than WaveGlow). Now, it is a very promising neural vocoder. We also recommend setting h = 16 for neural vocoding task, because it provides better fidelity of audio and its synthesis speed is only marginally slower than h = 8 with the help of convolution queue. **
>
> We will address your detailed comments in the following.
>
> - “The submission would have benefited from discussion about model complexity/expressivity and it's impact on MOS for WaveFlow, WaveNet and other approaches. ”
> * Many thanks for this great suggestion. We measure the complexity/expressivity of generative models in terms of likelihood.  We find a positive correlation between test likelihood and MOS score for these likelihood-based models (see Figure 3 in the updated version of the submission). In general, larger likelihood implies higher fidelity of speech. Indeed, we use the likelihood score as a performance indicator for designing WaveFlow. We will include more discussion in our final draft.
>
> - “1) lack of proper technical description of your model in sections 1 and 2 making reading sections 1,2,3,etc in order awkward. It seems the order should be 3,4,(5),1,2,(5). ”
> * Thank you for this nice suggestion. We have reorganized the related work section after the technical description of our model in the revision.
>
> -“2) complete omission of conditioning on text to be synthesised; anyone not familiar deeply with speech synthesis will wonder where does the text come in”
> * This is a good point.  We have added Section 3.3 to provide the details of conditioning on text.
>
> -“3) explicit statement of complexity for the operations involved using proper big-O notation; helps to avoid confusion about what do you mean by "parallel" (autoregressive WaveNet followed by parallel computation != parallel computation) ”
> * Thank you for your suggestion. We have explicitly stated the complexity for the operations using big-O notation in our revision.

---

### Official Review · AnonReviewer2 · 2019-10-25
**Official Blind Review #2**

**Rating:** 3

**Review:**

This paper re-organized the high dimensional 1-D raw waveform as 2-D matrix. This method simulated the autoregressive flow. Log-likelihood could be calculated in parallel. Autoregressive flow was only run on row dimension. The number of required parameters was desirable to synthesize high-fidelity speech with the speed faster than real time. Although this method could not achieve top one in ranking in every measurements, the resulting performance was still obtained with the best average results.

In general, this paper is clearly written, well organized and easy to follow. The authors carried out sufficient experiments and analyses, and proposed some rules of thumb to build a good model. On one hand, we may catch the contributions. But, on the other hand, the contributions were not clearly explained. The results were averaged but were not clearly explained.

The authors suggested to specify a bigger receptive field than the squeezed height. The property of getting better performance using deeper wavenet was "not" clearly explained and investigated. In the experiments, a small number of generative steps was considered. This is because short sequence based on autoregressive model was used.

This paper mentioned that using convolution queue could improve the synthesis speed. But, the synthesis speed has been fast enough because it is almost 15 times faster than real time. In practical applications, 100x faster is almost the same as 15x faster for humans. But, the task isn’t interacted with human. It is suggested to focuse on reducing the number of parameters or enhancing the log likelihood.

**Experience Assessment:**

I have published one or two papers in this area.

**Review Assessment: Checking Correctness Of Derivations And Theory:**

I assessed the sensibility of the derivations and theory.

**Review Assessment: Checking Correctness Of Experiments:**

I assessed the sensibility of the experiments.

**Review Assessment: Thoroughness In Paper Reading:**

I made a quick assessment of this paper.

---

> ### Author Response · Authors · 2019-11-14
> **Response to Official Blind Review #2**
>
> Many thanks for your review; the feedback is helpful to improve our paper.
>
> ** After the submission, we have made two improvements: 1) We find that the split & reverse operations for stacking multiple flows are more effective than reverse-only operations (see details in Section 3.4 in our revision). Our small-footprint WaveFlow now obtains larger test likelihood (see Table 3 in the revision) and improved speech fidelity (see Table 5 in the revision).   2) We implement convolution queue (Paine et al., 2016), which brings additional 3x to 5x speedup for WaveFlow models.  As a result, our small-footprint WaveFlow (5.91M parameters) can now generate 22.05kHz high-fidelity speech (MOS: 4.32) more than 40x faster than real-time (faster than WaveGlow), which provides a promising neural vocoder. We also recommend setting h = 16 for neural vocoding task, because it provides better speech fidelity and its synthesis speed is only marginally slower than h = 8 with the help of convolution queue. **
>
> We will address your comments in the following.
>
> - “On one hand, we may catch the contributions. But, on the other hand, the contributions were not clearly explained. The results were averaged but were not clearly explained.”
> * We can summarize our contributions in two points:
> (1) We propose a novel and unified framework for constructing likelihood-based generative models for raw audio, which includes previous approaches (WaveNet and WaveGlow) as special cases. We demonstrate the trade-off between memory footprint, generation speed and audio fidelity within the framework.
> (2) The resulting small WaveFlow is a compelling neural vocoder. In comparison with WaveGlow, it requires much fewer parameters (5.91M vs. 87.88M) to generate high fidelity speech (MOS: 4.32 vs. 4.34). Its synthesis speed is also slightly faster (42.60x vs. 34.69x).  In comparison with WaveNet, WaveFlow models are significantly faster at synthesis.
>
> - “The property of getting better performance using deeper wavenet was "not" clearly explained and investigated.”
> * We only test 30-layer WaveNet in the paper. We think this question was perhaps raised for flow-based models. In Table 4, we investigate WaveFlow with 6x8 = 48 and 8x8 = 64 layers (e.g., row-(l) vs. row-(m)), and WaveGlow with 6x8 = 48 and 12x8 = 96 layers (e.g., row-(e) vs. row-(f)), respectively. The models stacked with larger number of flows (i.e., deeper layers) consistently provide better likelihood. This property is also well known in normalizing flow literature (e.g., [1]). We have added details in Section 5.1.
>
> [1] Rezende and Mohamed. Variational inference with normalizing flows. ICML, 2015.
>
> - “This paper mentioned that using convolution queue could improve the synthesis speed. But, the synthesis speed has been fast enough because it is almost 15 times faster than real time. In practical applications, 100x faster is almost the same as 15x faster for humans. But, the task isn’t interacted with human. It is suggested to focus on reducing the number of parameters or enhancing the log likelihood.”
> * From human perceptual perspective, 15x faster and 40x faster (our new result)  than real-time has minor difference. However, the convolution queue removes redundant calculation at synthesis, which will also improve system throughput in practical applications. We do agree on that reducing parameters or enhancing the log likelihood is very important for flow-based models. The previously mentioned split & reverse operation is a new endeavor after the submission. Note that, there is still significant likelihood gap that has so far existed between autoregressive models and flow-based models [2]. Our proposed model can close the gap with larger squeezing factor h (e.g., h = 64 in Table 4), or increased model size.
>
> [2] Ho et al. Flow++: Improving Flow-Based Generative Models with Variational Dequantization and Architecture Design. ICML 2019.

---

### Public Comment · ~Joan_Serrà1 · 2019-09-27
**Audio flows paper**

Cool work!
Perhaps the authors could also be interested on our paper https://arxiv.org/abs/1906.00794

---

> ### Author Response · Authors · 2019-10-14
> **Thank you for your comment**
>
> Hi Joan, thank you for your interest in our work. You paper is also interesting and we will reference it in a future version of our paper.

---

### Public Comment · ~Yi_Ren2 · 2019-10-02
**Related work**

Quite interesting work!

And I would greatly appreciate it if you would cite our FastSpeech (NeurIPS 2019) paper which significantly speeds up the mel-spectrogram generation with non-autoregressive architecture.

FastSpeech: Fast, Robust and Controllable Text to Speech: https://arxiv.org/abs/1905.09263

---

> ### Author Response · Authors · 2019-10-14
> **Thanks for your comment**
>
> Hi Yi,  thanks for your interest in our work. Your work is also interesting and we will reference it in the final version of this paper.

---

### Author Response · Authors · 2019-10-14
**Further speed-up at synthesis with convolution queues (Paine et al., 2016)**

After this submission, we have implemented convolution queues (Paine et al., 2016) to cache the intermediate hiddens within WaveFlow for autoregressive inference over the height dimension. It can bring significant speed-up over vanilla implementation depending on the squeeze size h on height.  In particular, our small-footprint WaveFlow can generate 22.05kHz speech 47.61 times faster than real-time. The updated synthesis speed results are as follows:

Model                        residual channels      # param       synthesis speed
WaveFlow (h=8)                    64                        5.91M                 47.61x
WaveFlow (h=16)                  64                        5.91M                 42.60x
WaveFlow (h=8)                    96                      12.78M                 29.09x
WaveFlow (h=8)                  128                      22.25M                 23.44x
WaveFlow (h=8)                  256                      86.18M                   9.09x

---

### Author Response · Authors · 2020-04-23
**[DEPRECATED] This version is outdated**

We have released a new version of this paper on arXiv ( https://arxiv.org/abs/1912.01219 ), which is accepted to ICML 2020.

---

### Decision · Program_Chairs · 2019-12-19

**Decision:**

Reject

**Comment:**

The paper presented a unified framework for constructing likelihood-based generative models for raw audio. It demonstrated tradeoffs between memory footprint, generation speech and audio fidelity. The experimental justification with objective likelihood scores and subjective mean opinion scores are matching standard baselines. The main concern of this paper is the novelty and depth of the analysis. It could be much stronger if there're thorough analysis on the benefits and limitations of the unified approach and more insights on how to make the model much better.